# Genome-wide identification of neuronal activity-regulated genes in *Drosophila*

Xiao Chen[1,2], Reazur Rahman[1,2], Fang Guo[1,2], Michael Rosbash[1,2]*

[1]Howard Hughes Medical Institute, Brandeis University, Waltham, United States; [2]National Center for Behavioral Genomics, Department of Biology, Brandeis University, Waltham, United States

**Abstract** Activity-regulated genes (ARGs) are important for neuronal functions like long-term memory and are well-characterized in mammals but poorly studied in other model organisms like *Drosophila*. Here we stimulated fly neurons with different paradigms and identified ARGs using high-throughput sequencing from brains as well as from sorted neurons: they included a narrow set of circadian neurons as well as dopaminergic neurons. Surprisingly, many ARGs are specific to the stimulation paradigm and very specific to neuron type. In addition and unlike mammalian immediate early genes (IEGs), fly ARGs do not have short gene lengths and are less enriched for transcription factor function. Chromatin assays using ATAC-sequencing show that the transcription start sites (TSS) of ARGs do not change with neural firing but are already accessible prior to stimulation. Lastly based on binding site enrichment in ARGs, we identified transcription factor mediators of firing and created neuronal activity reporters.

*For correspondence: rosbash@brandeis.edu

**Competing interests:** The authors declare that no competing interests exist.

## Introduction

Upon stimulation, immediate-early genes (IEGs) are induced rapidly and transiently without de novo protein synthesis in mammalian neurons (*Fowler et al., 2011*). This activity-dependent gene regulation is important as it affects the ability of an animal to convert transient stimuli into long-term changes (*Huh et al., 2000*; *Kandel, 2001*; *Nestler, 2001*; *Spitzer et al., 2000*). Hundreds of IEGs, including those specific to tissues and stimulation paradigms, have been identified in mammals since the discovery of the first neuronal IEG, *c-fos*, by Greenberg and his colleagues in 1986 (*Greenberg et al., 1986*; *Herschman, 1991*; *Spiegel et al., 2014*). Expression of these IEGs are induced within an hour, usually with big amplitudes (over 10 fold), and many IEGs are shared between different types of neurons (*Spiegel et al., 2014*). Functionally, these IEGs are highly enriched for transcription factors, which subsequently trigger a secondary transcriptional response (*Spiegel et al., 2014*; *West and Greenberg, 2011*). The secondary response genes (SRGs) in contrast take longer to induce (a typical assay is 6 hr post-stimulation), are involved in many different processes, are more neuron-specific and function at least in part to promote neuron survival, dendritic morphogenesis and regulate synapse formation (*Bloodgood et al., 2013*; *Hong et al., 2008*; *Lin et al., 2008*).

Not all IEGs respond to all types of stimulations, even the most robust ones like *c-fos*, *Egr1* and *Arc* (*Bepari et al., 2012*; *Fields et al., 1997*). For example, different stimulation paradigm-dependent Ca$^{2+}$ entry routes initiate different downstream pathways and lead to induction of distinct IEGs (*West and Greenberg, 2011*). Stimulations other than neural firing like growth factors also induce IEG expression, many of which are the same as those induced with neural firing (*Jones et al., 1988*; *Tullai et al., 2007*). Depending on the induction dynamics, Tullai *et al.* divided the platelet-derived growth factor (PDGF)-induced IEGs in human T98G glioblastoma cells into primary response genes (PRGs) and delayed response genes (DRGs). Dramatic differences were shown between the two

categories in terms of their functions, gene lengths, chromatin accessibility and RNA polymerase II occupancy at promoter regions (*Fowler et al., 2011*; *Kim et al., 2010*; *Tullai et al., 2007*). PRGs are usually optimized for rapid induction (such as shorter gene length and more permissive chromatin at promoters), whereas DRGs are not different from other genes in the genome. Notably, PRGs and DRGs are induced independent of protein-synthesis ('cycloheximide-insensitive'), whereas SRGs are protein-synthesis dependent ('cycloheximide-sensitive').

IEGs, or activity-regulated genes (ARGs; they are defined here as induced rapidly with neuronal activity, i.e., mostly within an hour, but without regard to de novo protein synthesis) are poorly defined in organisms other than mammals. Only three genes to date have been identified as responding to increased neural activity in insects: *kakusei* (in honey bee), *stripe* (abbreviated as *sr*, in honey bee) and *hr38* (in silkmoth and the fruit fly *Drosophila melanogaster*) (*Fujita et al., 2013*; *Kiya et al., 2007*; *Lutz and Robinson, 2013*). Moreover, only one genome-wide study using microarrays had been performed more than a decade ago to identify seizure-induced ARGs from fly heads (*Guan et al., 2005*). More high-throughput studies on *Drosophila* ARGs are therefore needed, not only for identification and to provide mechanistic insight but also to design new tools to serve as indicators of neuronal activity.

Here, we identified ARGs in a genome-wide manner, in fly brains as well as in sorted neurons; they included dopaminergic neurons (DA) and a subset of circadian-related neurons (PDF+ neurons). Fly ARGs vary with the individual stimulation paradigm and are surprisingly cell type-specific. Fly ARGs are also more functionally diverse and have longer gene lengths compared to mammalian ARGs. Chromatin at transcription start sites (TSS) of fly ARGs is more accessible at baseline than other expressed genes but does not change with stimulation. Lastly, we used bioinformatics to identify key transcription factors that mediate ARG activation. Based on these factors, we generated novel luciferase reporters for in vivo monitoring of neuronal firing.

## Results

### Genome-wide identification of firing-induced ARGs in fly brains

To identify ARGs in fly neurons in a genome-wide manner, the pan-neuronal driver *Elav-GAL4* was used to drive expression of *UAS-ChR2-XXL*, to artificially fire neurons by illuminating flies with a blue LED (*Dawydow et al., 2014*). A 30 s 10 Hz LED exposure was sufficient to induce a uniform seizure within seconds, and all flies were able to recover within 15 min. Fly brains were dissected either before (0 min), or 15, 30, 60 min after stimulation (*Figure 1A*). Flies expressing only *Elav-GAL4* were illuminated in parallel and used as a control strain. RNA was then extracted from these samples and made into mRNA libraries for deep sequencing.

96 genes show significant increases in 60 min with p-value < 0.01 of both exact test and general linear model (GLM) in *UAS-ChR2-XXL* flies (N = 3 biological replicas); very few genes show significant decreases (*Figure 1B*). Ranking #1 and #2 are the previously identified insect IEGs *hr38* and *sr*, with a 90- and a 10- fold increase respectively, suggesting the stimulation paradigm is sufficient to fire neurons in the brain (*Figure 1C*, *Figure 1—source data 1*). The 30 s LED exposure affected many fewer genes in control flies: four were up-regulated, and three down-regulated with p-values<0.01 by both statistical tests (*Figure 1—figure supplement 1A*, N = 3). 3 of the four up-regulated genes, *CG13055, hsp23* and *hsromega,* had induction amplitudes statistically indistinguishable from those of the *UAS-ChR2-XXL* flies and are therefore labeled as light-induced genes in *Figure 1—source data 1* (*Figure 1—figure supplement 1B*). However, they were not removed from the ARG list because light can induce neuronal firing and some of these genes are also induced by other stimulation paradigms (see below).

Gene Ontology (GO) analysis using GOrilla (http://cbl-gorilla.cs.technion.ac.il/) on the 96 ARG genes shows that polII transcription factor activity is the most enriched function with a P-value of 1.29E-4 computed according to the mHG or HG model, followed by steroid hormone receptor activity and MAP kinase activity (*Figure 1D*). Several of these induced transcription factors encode proteins with high sequence identity to well-known mammalian IEGs: HR38 shows 67% identity to nuclear receptor subfamily four group A member 2 (NR4A2); SR has 82% identity to early growth response protein 3 (EGR3); CBT is 61% identical to Krueppel-like factor 11 (KLF11). Taken together,

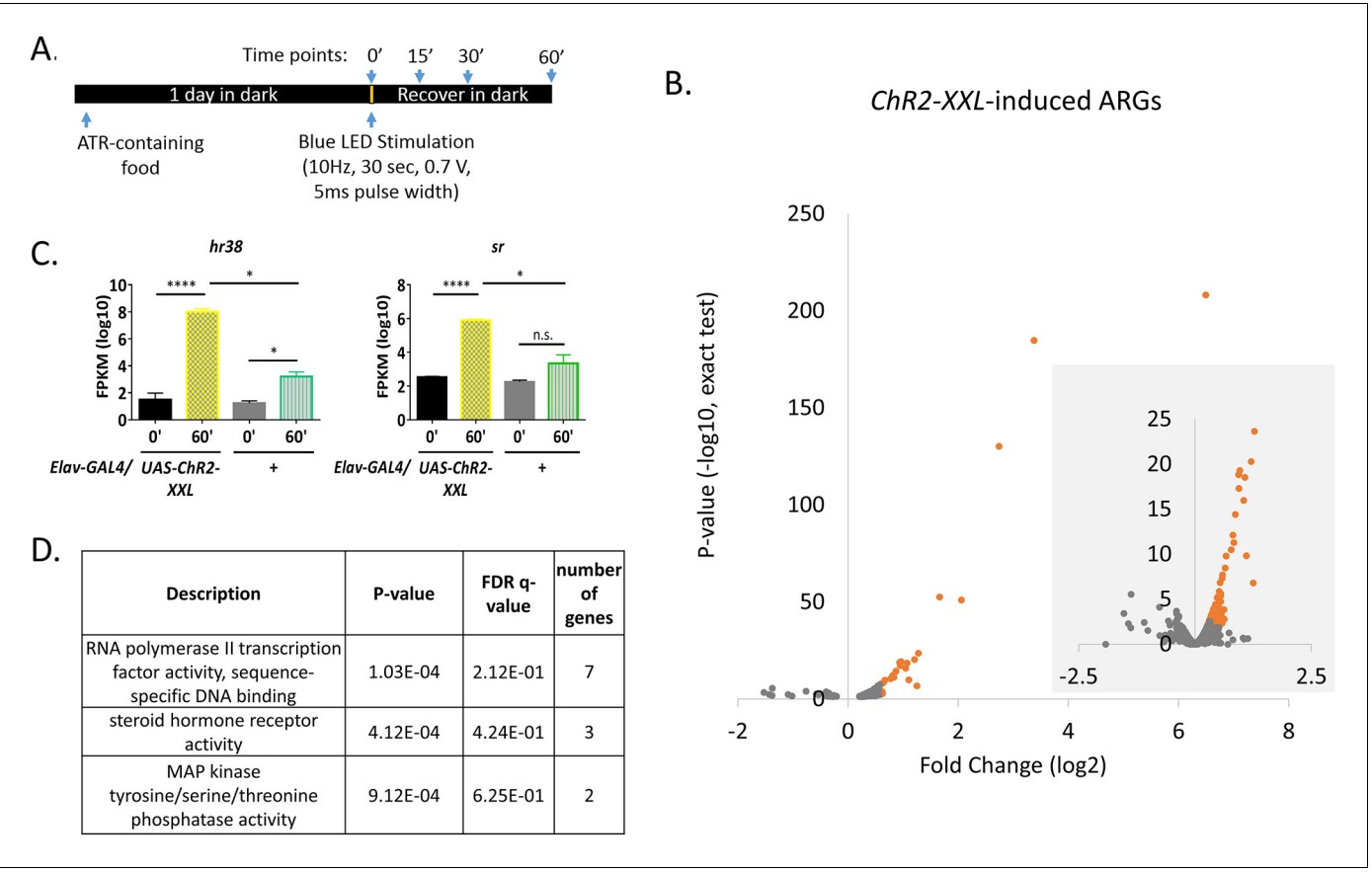

**Figure 1.** High-throughput sequencing and optogenetics reveal ARGs in *Drosophila*. (**A**) Stimulation paradigm for *Elav-GAL4;UAS-ChR2-XXL* flies. (**B**) Volcano plots of individual genes (FPKM > 0) with log2 fold changes (x-axis) against p-value (y-axis, exact test). Genes significantly induced (p-value < 0.01 for both exact test and GLM) are in red. The smaller area with grey background is the zoom-in view with larger scales. N = 3 biological replicates. (**C**) Gene expression of *hr38* and *sr* in *Elav-GAL4;UAS-ChR2-XXL* and *Elav-GAL4/+* (control) flies 0 or 60 min after LED stimulation. N = 3 biological replicates, error bars represent ±SEM, n.s. represents non-significant. *p<0.05, ****p<0.0001, exact test. (**D**) Gene ontology analysis on *ChR2-XXL*-induced ARGs. P-value is the enrichment p-value, not corrected for multiple testing. FDR q-value is the corrected p-value for multiple testing using the Benjamini and Hochberg (1995) method (*Eden et al., 2009*; *Eden et al., 2007*). Number of genes indicates the number of ARGs with the specified functions.

The following source data and figure supplement are available for figure 1:

**Source data 1.** *ChR2-XXL*-induced ARGs in fly brains.

**Figure supplement 1.** Light-induced gene expression.

firing neurons with *ChR2-XXL* identified known as well as novel ARGs in flies, and the enrichment of transcription factors in ARGs suggests some mechanistic similarity between flies and mammals.

## Gene sizes of ARGs correlate with induction rates

To assess the kinetics of ARG induction and to provide a global view, a heat map across time points was generated (*Figure 2A*). The levels of most ARGs continue to increase until the last time point assayed (60 min); only a few transcripts like *cbt* (2.6 kb) reach maximum expression levels at 30 min (*Figure 2B*). Since shorter gene length enables the rapid induction of IEGs in mammals (*Tullai et al., 2007*), we examined the gene lengths of fly ARGs. Surprisingly, the average ARG gene is longer than that of all annotated genes as well as of all genes expressed in neurons (*Figure 2C*, gene annotation obtained from FlyBase (ftp://ftp.flybase.net/genomes/dmel/dmel_r5.2_FB2007_01/fasta)). This is true even for some of the most robust ARGs; *hr38* is ~31 kb long, *sr* spans ~ 11 kb, *CG8910* is ~21

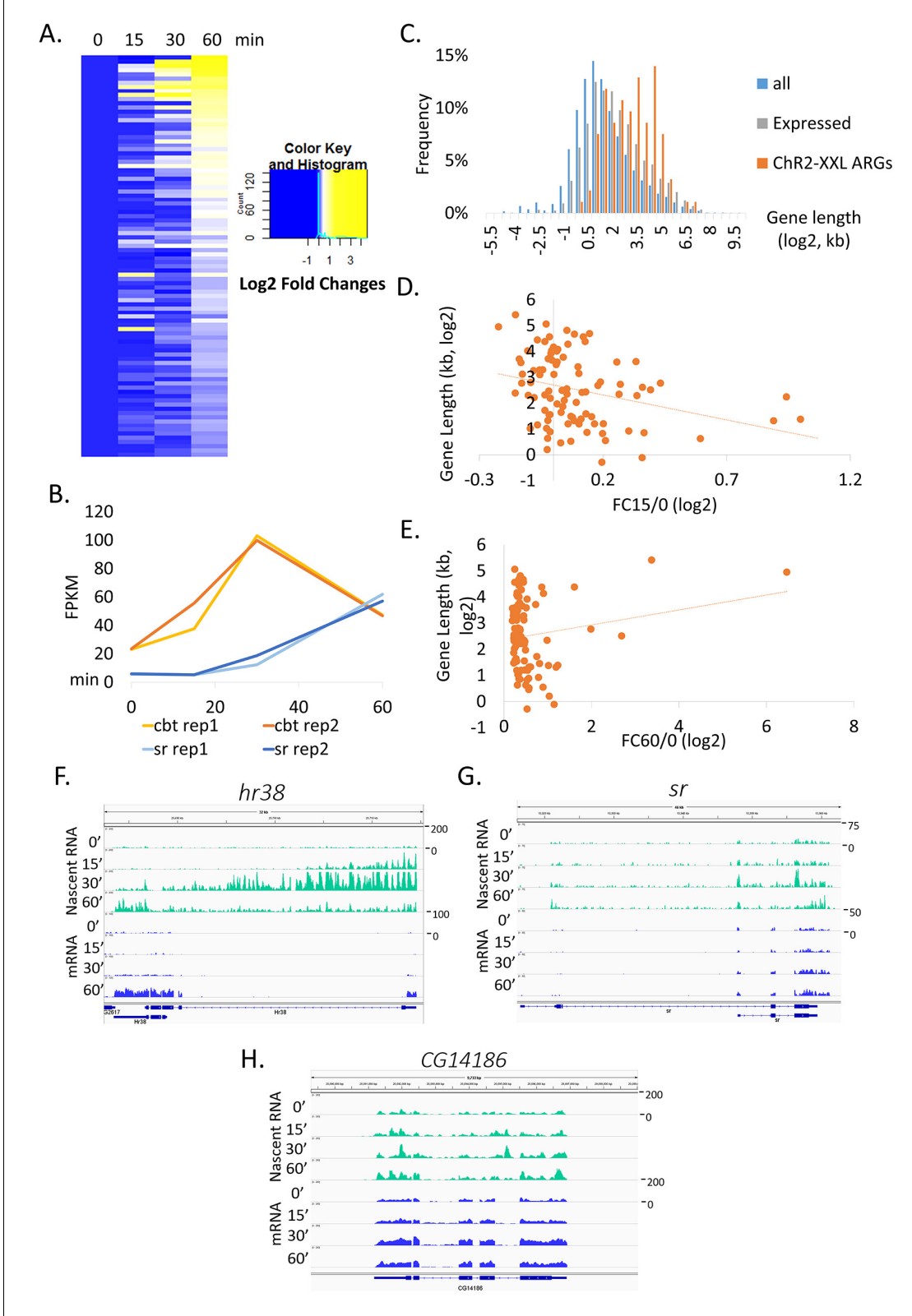

**Figure 2.** Gene sizes of ARGs correlate with induction rates. (**A**) Heat map of *ChR2-XXL*-induced ARG expression from 0, 15, 30 to 60 min (normalized to 0 min expression) post-LED stimultaion. Each row represents one gene. Data are averages of 2 biological replicates. (**B**) Gene induction profiles of *cbt* (gene length = 2.6 kb, red) and *sr* (gene length = 11 kb, blue) with LED stimulation in *Elav-GAL4;UAS-ChR2-XXL* flies (N = 2 biological replicates, abbreviated as rep1/2). (**C**) Gene length distribution of all annotated genes in the genome (blue), genes expressed in brains (gray, FPKM > 0 at

*Figure 2 continued on next page*

*Figure 2 continued*

baseline, FPKM > 10 with stimulation), and *ChR2-XXL*-induced ARGs (red). Only the longest isoform of each gene is plotted. (**D**) Correlation of *ChR2-XXL*-induced ARGs gene length (y-axis) and fold changes of 15 min compared to 0 min (abbreviated as FC15/0, x-axis). (**E**) Correlation of *ChR2-XXL*-induced ARGs gene length (y-axis) and fold changes of 60 min compared to 0 min (abbreviated as FC60/0, x-axis). (**F–H**) Expression profiles of *hr38, sr,* and *CG14186* from fly heads 0, 15, 30 and 60 min after LED stimulation. Nascent RNA levels are shown in green and mRNA levels in blue. Numbers to the right of the expression graphs indicate data range.

The following figure supplement is available for figure 2:

**Figure supplement 1.** Gene structure of some long ARGs induced by *ChR2-XXL*, including *hr38, sr, CG8910,* and *CG11221*.

kb and *CG11221* is ~13 kb (*Figure 2—figure supplement 1*). In contrast, the canonical mammalian IEGs *c-fos* and *c-jun* are only ~ 3.4 kb and ~3.2 kb respectively. This is despite the fact that the average gene size of flies (~9.3 kb) is considerably smaller than that of mammals (~47 kb for mouse).

To test if gene length affects ARG induction rates, we compared the fold changes of gene expression between 0 and 15 min (FC15/0) relative to gene lengths. There is a weak anti-correlation, suggesting that longer genes are indeed more slowly induced on average (*Figure 2D*). This is not because overall induction amplitudes are smaller for longer genes, since the correlation disappears if fold changes between 0 and 60 min (FC60/0) are compared to gene lengths (*Figure 2E*).

To gain more insight into these dynamics, nascent RNA and mRNA libraries were constructed in parallel from fly heads using the same stimulation paradigm. Heads rather than brains were used because of technical considerations: the much larger amounts of starting material required for nascent RNA preparation are difficult to obtain with brain dissections.

Although *hr38* and *sr* mRNA levels peak at 60 min with little or no changes evident at 15 min and even at 30 min (*Figure 2F,G*), the nascent RNA data indicate that the enhanced transcription of both genes is obvious at the first 15 min time point and peaks at 30 min, 30 min before the peak time of mRNA (*Figure 2F,G*). For shorter genes such as *CG14186* (5.7 kb), the stimulation-induced increases are smaller at the nascent RNA level, presumably because much less time is required for transcription elongation and polyadenylation. More importantly and unlike the longer genes, there is no delay between the nascent and mRNA peak times for the shorter genes (*Figure 2H*). The data taken together indicate that the long gene lengths of some prominent ARGs reduce the mRNA induction rates due to the time required for nascent events like transcriptional elongation.

## ARGs induced by two additional stimulation paradigms

To test whether these ARGs are universal and respond to different stimulation regimens, we employed two additional paradigms: *dTrpA1*-mediated firing and high KCl-induced depolarization (*Figure 3A*). *UAS-dTrpA1* was overexpressed under the control of *Elav-GAL4*, and flies were stimulated in vivo by shifting the temperature from 18°C to 29°C to stimulate the temperature-sensitive *dTrpA1* cation channel. High KCl stimulation was done ex vivo by incubating dissected brains in 90 mM KCl. In this case, the voltage sensor ArcLight was used to confirm successful depolarization (*Video 1*, *Figure 3—figure supplement 1*) (*Cao et al., 2013*).

Hundreds of genes are induced with the *dTrpA1* paradigm but many are also robustly heat-induced, i.e., they are also induced in control flies expressing only *Elav-GAL4*. Nonetheless, 97 genes including *hr38* and *sr* were induced by *dTrpA1* at 60 min but not induced or induced much less robustly in the control flies (*Figure 3B*, *Figure 3—source data 1*). With KCl treatment, only 38 genes were significantly induced in 60 min, and most of them are stress-induced such as heat shock genes (*Figure 3—source data 2*). *Hr38* and *sr* were also significantly induced by KCl but with much smaller increases (2.5 and 1.5 fold, respectively) compared to the other two paradigms (*Figure 1—source data 1*, *Figure 3—source data 1*). To accommodate genes that might be induced more slowly with the KCl paradigm, an extra 90 min time point was assayed (three biological replicas). Because many more ARGs (107 genes in total) were significantly induced and had bigger induction amplitudes at 90 min compared to 60 min (*Figure 3—source data 3*, *Figure 3C*), this 90 min time point was compared with the other two induction methods.

Only 12 ARGs are shared by all the three paradigms (including *hr38* and *sr*; *Figure 3D,E*, *Figure 3—source data 4*). To verify this conclusion, qPCR was used on 6 of these shared genes, and

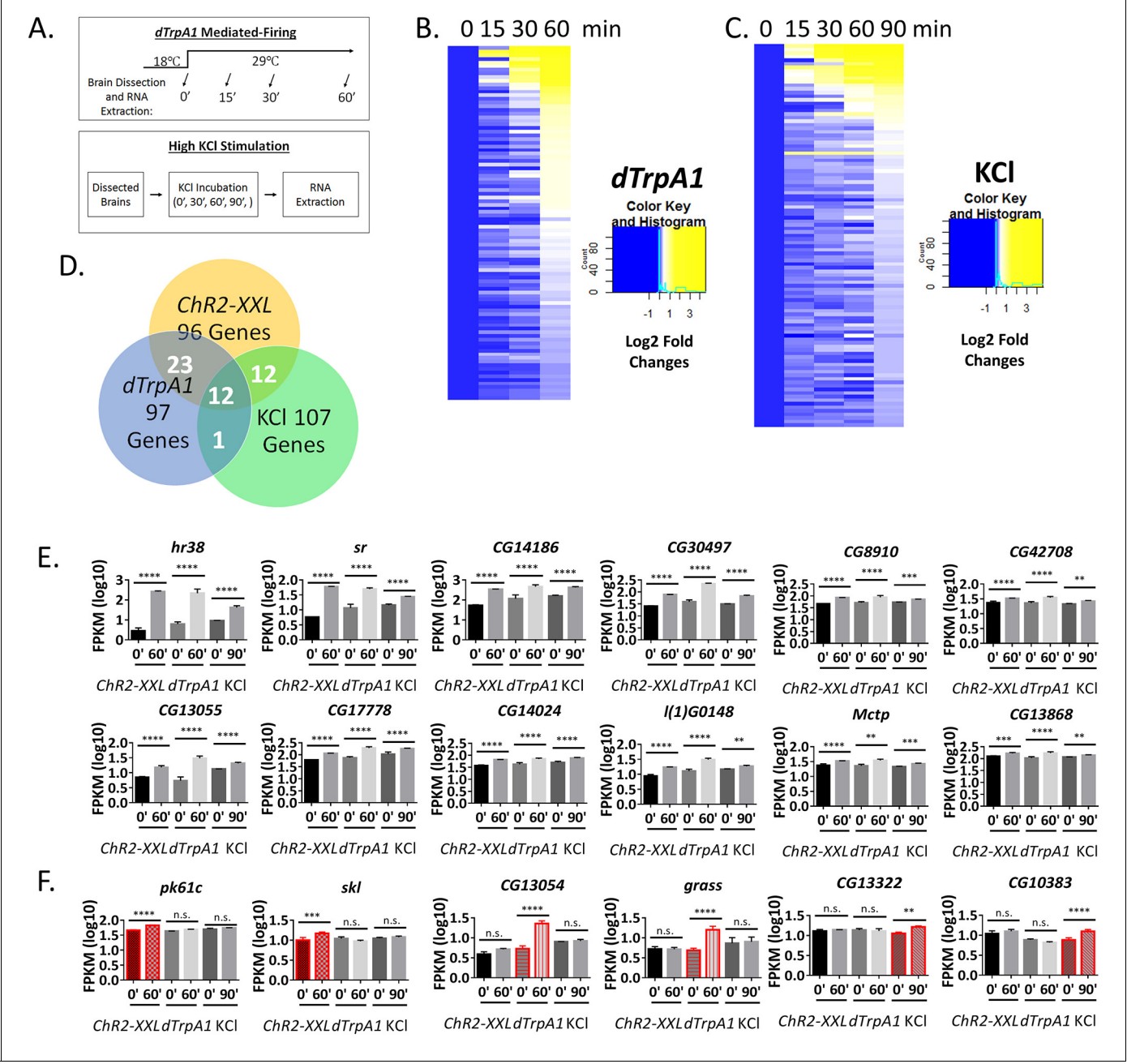

**Figure 3.** ARGs induced by two additional stimulation paradigms. (**A**) Schemes of two additional stimulation paradigms: *dTrpA1* and KCl. (**B**) Heat map of *dTrpA1*-induced ARG expression with 0, 15, 30 and 60 min temperature shift (normalized to 0 min expression). N = 3 biological replicates. (**C**) Heat map of KCl-inducd ARG expression with 0, 15, 30, 60 and 90 min KCl incubation (normalized to 0 min expression). N = 3 biological replicates. (**D**) Overlap of ARGs among the three different stimulation paradigms: *ChR2-XXL, dTrpA1* and KCl (90 min). (**E**) Gene expression of the 12 shared ARGs induced by all three stimulation paradigms. N = 3 biological replicates, error bars represent ±SEM, **p<0.01, ***p<0.001, ****p<0.0001, exact test. (**F**) Gene expression of paradigm-specific ARGs: *pk61c* and *skl* are specific to the *ChR2-XXL* paradigm; *CG13054* and *grasss* are specific to the *dTrpA1* paradigm; and *CG13322* and *CG10383* are specific to the KCl paradigm. N = 3 biological replicates, error bars represent ±SEM, n.s. represents non-significant, **p<0.01, ***p<0.001, ****p<0.0001, exact test.

The following source data and figure supplements are available for figure 3:

**Source data 1.** *dTrpA1*-induced ARGs in fly brains.

**Source data 2.** KCl-induced ARGs in 60 min in fly brains.

*Figure 3 continued on next page*

*Figure 3 continued*

**Source data 3.** KCl-induced ARGs in 90 min in fly brains.

**Source data 4.** Overlapped ARGs in brains.

**Figure supplement 1.** Fluorescence changes of ArcLight indicate successful depolarization of neurons with KCl stimulation.

**Figure supplement 2.** qPCR verification of high-throughput RNA sequencing results.

**Figure supplement 3.** qPCR quantification of paradigm-specific ARGs in *ChR2-XXL* flies with a 30 min LED stimulation followed by another 30 min recovery.

**Figure supplement 4.** qPCR quantification of ARGs induction under ex vivo condition.

the results of all six are qualitatively consistent with the high-throughput RNA sequencing results (***Figure 3—figure supplement 2A***).

The overlap between the three different pairwise combinations is larger: 13 genes are shared between *dTrpA1* (13% of *dTrpA1*-induced ARGs) and KCl (12% of KCl-induced ARGs), and 24 genes between *ChR2-XXL* (25% of *ChR2-XXL*-induced ARGs) and KCl (22% of KCl-induced ARGs). The largest number (35) and fraction of genes are shared between *ChR2-XXL* (36% of *ChR2-XXL*-induced ARGs) and *dTrpA1* (36% of *dTrpA1*-induced ARGs). These data indicate that many ARGs respond to firing independent of the stimulation paradigm (***Figure 3D***, ***Figure 3—source data 4***).

There are however some genes that are specific to the stimulation paradigm, consistent with what has been observed in mammals (***Bepari et al., 2012***; ***Fields et al., 1997***). For example, *pk61c* and *skl* are induced only by *ChR2-XXL; CG13054*, whereas *grass* is induced only by *dTrpA1; CG13322* and *CG10383* are only induced by KCl (***Figure 3F***). These stimulation paradigm-specific ARGs and six others were verified by qPCR (***Figure 3—figure supplement 2B***), with conclusions qualitatively consistent with high-throughput RNA sequencing results.

Stronger stimulation paradigms, for example longer LED exposure for *ChR2-XXL* flies, bigger temperature shift for *dTrpA1* flies or incubation with higher KCl concentration, may further increase the overlap among paradigms. To test this, the LED exposure of the *ChR2-XXL* flies was extended from 30 s to 30 min. 2 out of 8 paradigm-specific ARGs that were negative with the 30 s exposure were positive with the 30 min stimulation, but the other six remained negative (***Figure 3—figure supplement 3***). In addition, we tested by qPCR whether some of the KCl-specific ARGs are due to the ex vivo condition only used for the KCl-stimulation: 4 out of 4 tested are not induced by the other two stimulation paradigms performed under the same ex vivo condition, i.e., exposure of dissected brains to the LED or to temperature shift (***Figure 3—figure supplement 4A***). In contrast, four common ARGs were stimulated ex vivo by *ChR2-XXL* and *dTrpA1* in 60 min to levels quite comparable to those from the KCl paradigm at 90 min, suggesting that the different results are due to the different stimulation paradigms rather than ex vivo vs. in vivo stimulation (***Figure 3—figure supplement 4B***).

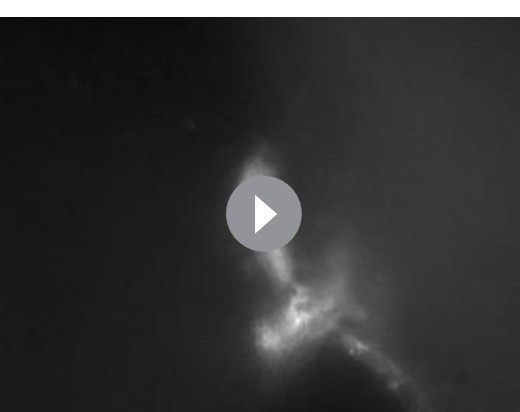

**Video 1.** S1 Live imaging of Arclight in PDF+ neurons in response to KCl treatment. To show KCl treatment can successfully depolarize neurons, Arclight was artificially expressed in PDF+ neurons (*Pdf-GAL4;UAS-ArcLight*). Arclight signal was monitored with fluorescent microscope while KCl was perfused into the chamber containing a dissected brain. Quantification of fluorescent intensity is shown in ***Figure 3—figure supplement 1***.

The results taken together suggest that most paradigm-specific ARGs will remain responsive to only one type of stimulation despite the fact that stronger stimulation will somewhat increase the overlap among the three paradigms, e.g., the distinction between the paradigms might be quantitative as well as qualitative. In any case, the data taken together significantly extend the list of *bona fide* ARGs in flies.

## Transcriptome profiling in DA and PDF+ neurons

Fly brains contain many types of neurons with distinct properties as well as glia and other tissues (The *Interactive* Fly, http://www.sdbonline.org/). Transcript changes in specific neurons may therefore be invisible with assays from brains and heads, especially if there is significant cell-type specificity in the transcriptional response to neuronal firing. To address this possibility, we performed mRNA deep sequencing on sorted green fluorescent protein (GFP) labeled DA and PDF+ neurons expressing *dTrpA1* with or without a temperature shift, namely, the same *dTrpA1* paradigm used for brains. DA and PDF+ neurons are two distinct populations and serve different functions in flies. Each brain has about ~120 DA neurons and ~16 PDF+ neurons. Flies expressing only the *GAL4* drivers were used as a control for heat-induced genes.

Despite much less starting material than brains, expression profiles from both cell types are well correlated between replicates ($R^2 > 0.9$ between replicates) (*Figure 4A,B*). In contrast, baseline expression profiles among cells and tissues (brains, PDF+ and DA neurons) are distinct as expected (*Figure 4—figure supplement 1*). Previously identified cell-type enriched transcripts were used as markers to ensure that the correct neurons were collected. The housekeeping gene Act5C is expressed ubiquitously in brains, DA and PDF+ neurons (*Figure 4C*). The DA neuron-enriched transcript *ple* (encodes tyrosine hydroxylase) and the PDF+ neuron-specific transcript *pdf* are both highly

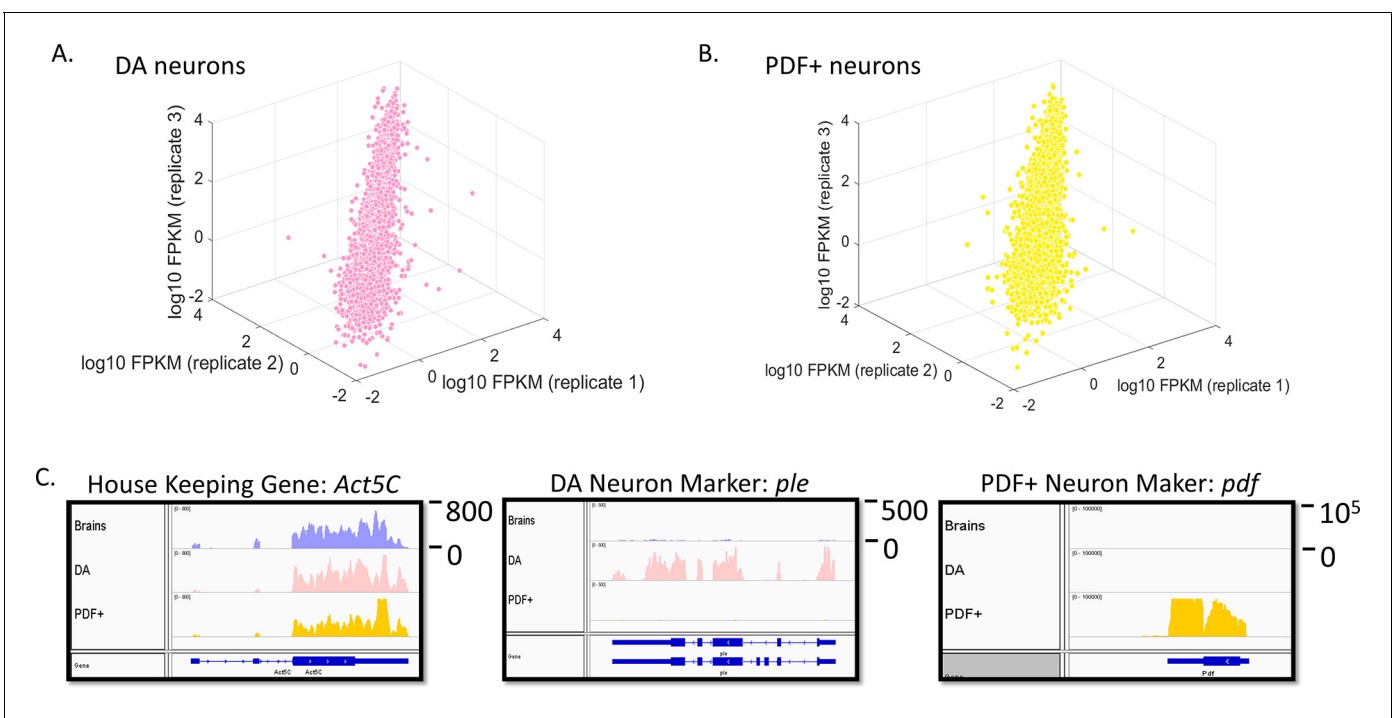

**Figure 4.** Transcriptome profiling in sorted DA and PDF+ neurons. (**A**) Gene expression (log10 FPKM) plot among three biological replicates (baseline levels) in DA neurons. Only genes with FPKM > 0.01 are plotted. (**B**) Gene expression (log10 FPKM) plot among three biological replicates (baseline levels) in PDF+ neurons. Only genes with FPKM > 0.01 are plotted. (**C**) Expression levels of the house-keeping gene (*act5c*) and marker genes (*ple* specific for DA neurons and *pdf* specific for PDF+ neurons) in all three tissues. Numbers to the right of the expression graphs represent data range.

The following figure supplement is available for figure 4:

**Figure supplement 1.** Transcriptomes of brains, DA and PDF+ neurons are very distinct.

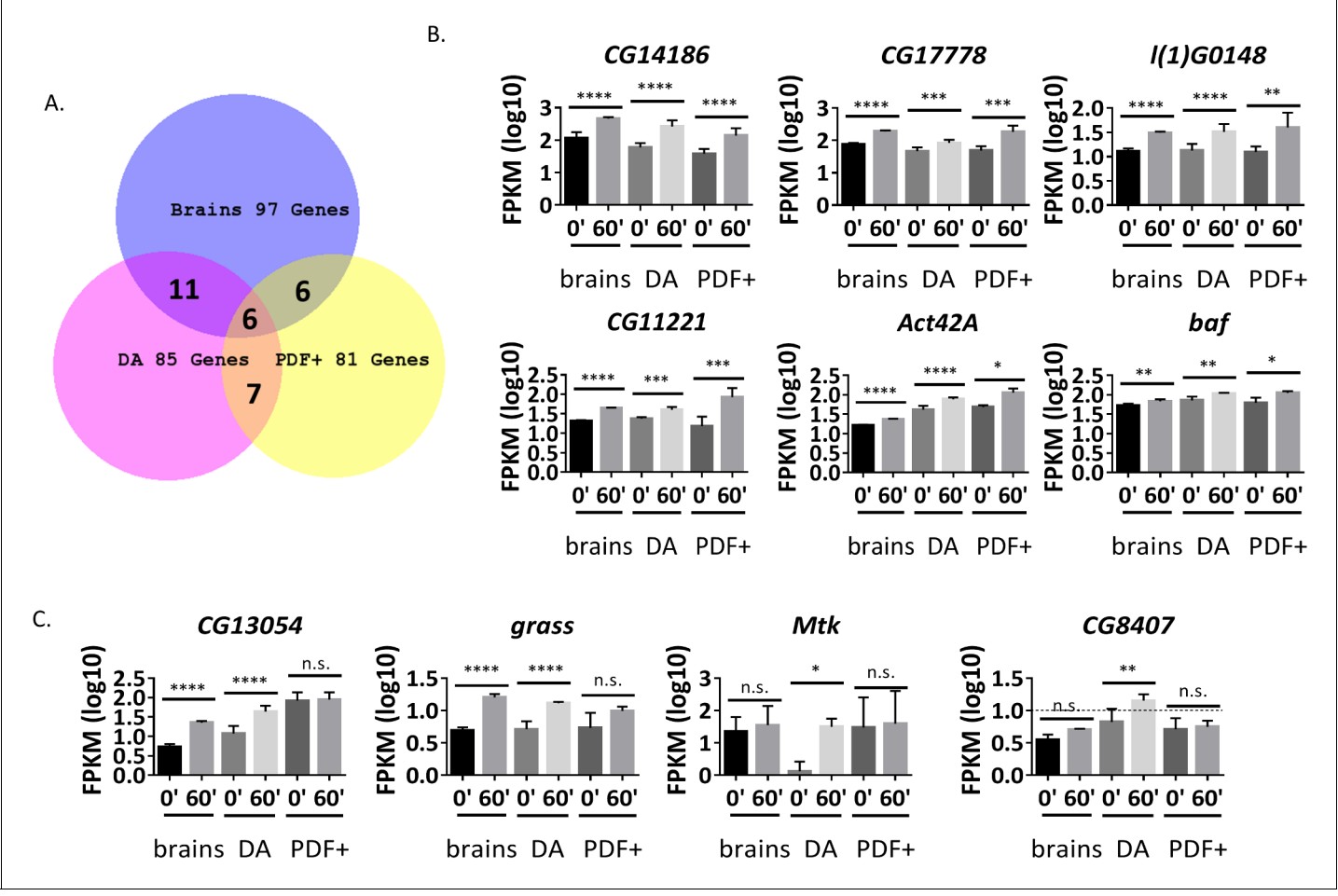

**Figure 5.** Many ARGs are tissue-specifically regulated in response to firing. (**A**) Overlap of ARGs in different tissues. (**B**) Gene expression of shared ARGs induced in all tissue types. N = 3 biological replicates, error bars represent ±SEM, *p<0.05, **p<0.01, ***p<0.001, ****p<0.0001, exact test. (**C**) Gene expression of tissue-specific ARGs. N = 3 biological replicates, error bars represent ±SEM, n.s. represents non-significant, *p<0.05, **p<0.01, ***p<0.001, ****p<0.0001, exact test.

The following source data and figure supplements are available for figure 5:

**Source data 1.** ARGs induced in DA neurons.
**Source data 2.** ARGs induced in PDF+ neurons.
**Source data 3.** Overlapped ARGs in different tissue types.
**Figure supplement 1.** Not all common ARGs in brains are induced in DA and PDF+ neurons.
**Figure supplement 2.** Big overlap among heat-induced genes in different tissues.

enriched only in the corresponding neuron types, indicating the correct cell types were collected (*Figure 4C*) (*Budnik and White, 1987*; *Helfrich-Förster, 1997*).

## There are many neuron-specific ARGs

85 candidate ARGs in DA neurons and 81 in PDF+ neurons are significantly induced in *dTrpA1* flies but not in control flies (*Figure 5A*, *Figure 5—source data 1* and *2*). There was also very limited overlap between the three sets of ARGs: only six genes are in common between all three sources, brains, DA and PDF+ neurons (*Figure 5A,B*); 11 additional genes are shared between brains and DA

neurons; seven between DA and PDF+; and six between brains and PDF+ neurons (*Figure 5A*, *Figure 5—source data 3*). Notably, half of the six common genes are among the 12 genes induced by all three stimulation paradigms in brains (*CG14186, CG17778* and *l(1)G0148, Figure 3E*), suggesting that these genes respond to firing in many and perhaps most fly brain neurons (*Figure 3—source data 4*). However, the other nine stimulation-independent brain ARGs are not universally induced, further indicating considerable cell type-specificity of firing-induced gene expression (*Figure 5—figure supplement 1*). Previous work on *hr38* using in situ hybridization is consistent with this interpretation (*Fujita et al., 2013*).

Indeed, these data indicate that most ARGs are cell type-specific (*Figure 5A*). For example, *CG13054* and *grass* are only significantly induced in brains and DA but not PDF+ neurons; *Mtk* and *CG8407* are only significantly induced in DA neurons; *CG9313* and *DAT* are exclusively up-regulated in PDF+ neurons (*Figure 5C*). There are also genes like *JhI-21* and *CG10863*, which show changes in sorted DA and PDF+ neurons but smaller or no changes in brains; this suggests masking/dilution effects in brains (*Figure 5C*). In contrast, almost complete overlap was observed for heat-induced genes (*Figure 5—figure supplement 2*). In sum, a preponderance of cell-type specific ARGs suggests that they endow their neurons with specific functions and lead to different firing-induced consequences.

## Promoter regions of ARGs are at permissive state prior to stimulation

Chromatin structure plays an important role in regulating gene expression, and promoters (TSS) of mammalian IEGs are usually located in open chromatin (*Fowler et al., 2011*). To address the chromatin states of fly ARG TSSs, we performed ATAC-seq in fly brains expressing *Elav-GAL4; UAS-dTrpA1* or its sibling control strain expressing *Elav-GAL4; CyO* before or 30 min after a temperature shift.

The TSS and its surrounding region (±250 bp) of all annotated fly genes are more accessible than other regions in the genome as expected (*Figure 6—figure supplement 1*). This region of ARGs also shows enhanced accessibility but does not detectably change with firing despite the increase in transcription (*Figure 6A*). Even the TSS accessibility of top-ranked ARGs like *hr38* (rank #1) and *CG7995* (rank #6) shows no differences with firing (*Figure 6B*, *Figure 6—figure supplement 2*). This is in contrast to the TSS of heat shock protein genes (HSPs), which possesses less accessible chromatin than that of ARGs at baseline but opens substantially by 30 min in both the *dTrpA1* and control flies in response to heat (*Figure 6C*). In fact, the whole gene body of heat shock genes opens with heat, consistent with what was shown previously at the *hsp70* locus (*Petesch and Lis, 2008*).

Despite the lack of a firing-induced change in ARG chromatin TSSs (*Figure 6A*), they are notably more accessible prior to stimulation compared to all expressed, annotated genes (*Figure 6D*). Since baseline ATAC-seq profiles are robust and minimally affected by genetic background (unpublished data from Rosbash lab), baseline chromatin profiles from ARGs induced by the other paradigms were pooled for analysis. These ARGs, like the *dTrpA1*-induced ARGs, also show higher accessibility than the rest of the genome (*Figure 6D*). This is unlikely due to higher baseline expression levels; only KCl-induced ARGs show significantly higher baseline expression levels despite showing intermediate chromatin accessibility of the three firing paradigms (*Figure 6—figure supplements 3*, *Figure 6D*). In conclusion, ARG TSS regions are generally more accessible at baseline, presumably to allow rapid transcriptional induction upon stimulation.

## Generation of luciferase reporters for neuronal activity

Since little was previously known about fly ARGs, no useful neuronal activity reporters comparable to c-fos have been constructed. We therefore used the information in this study to generate several luciferase reporters for monitoring neuronal activity in living flies. We used the online motif enrichment tool (http://veda.cs.uiuc.edu/cgi-bin/MET/interface.pl) to identify transcription factor binding sites in either the 1 kb or 5 kb upstream region of *ChR2-XXL*-induced ARGs; expressed genes (FPKM > 0 at baseline and FPKM > 10 with stimulation) were used as a null set for background subtraction. *Lola* (significance = 4.13E-07), *eip78C* (significance = 5.92E-05), *relish* (*rel*, significance = 1.78E-04), *broad* (*br*, significance = 6.62E-02, insignificant) and *cf2* (significance = 0.43, insignificant) were chosen to represent different levels of enrichment ranking, from high to low (*Figure 7—source data 1* and *2*). Three binding sites of each transcription factor were multimerized and

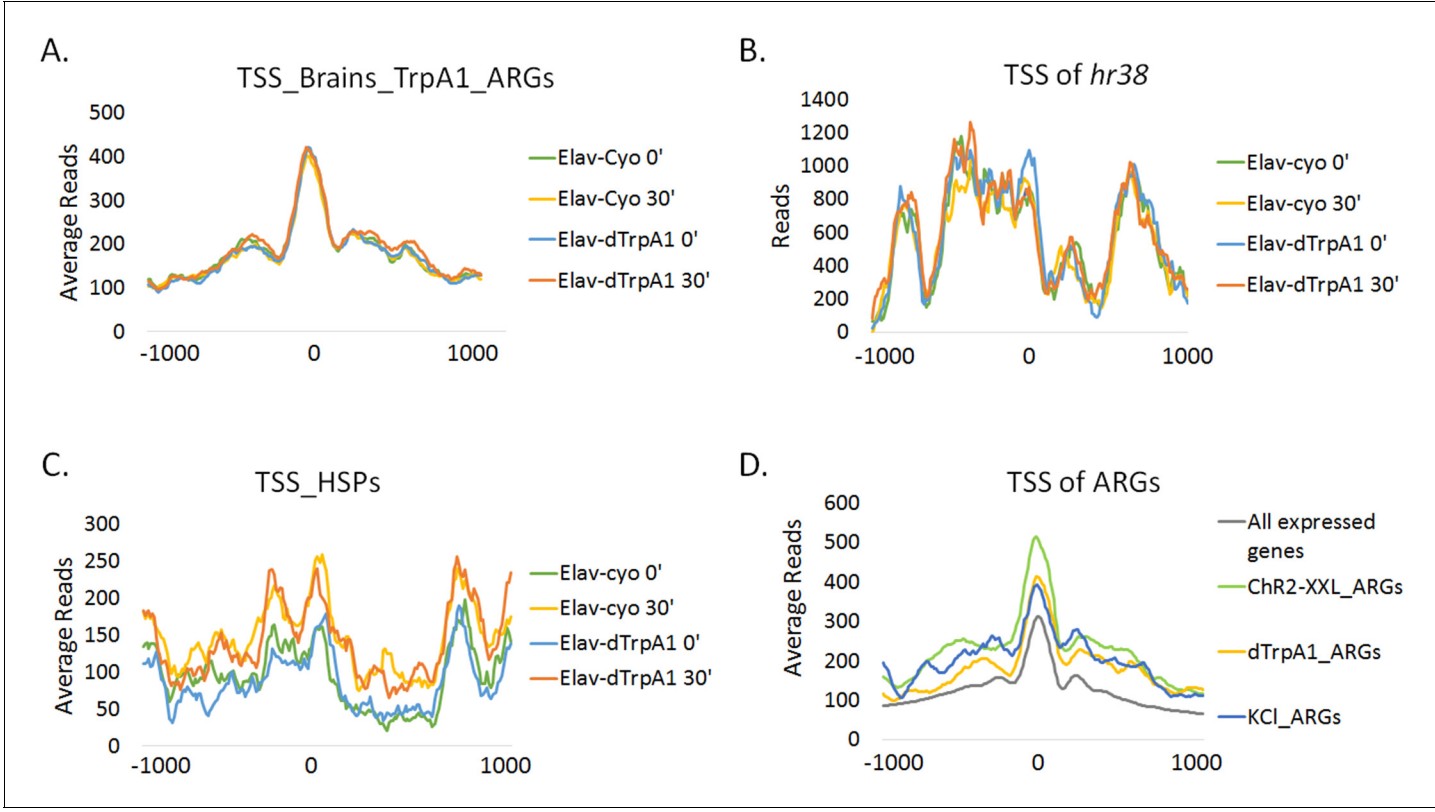

**Figure 6.** Promoter regions of ARGs are at permissive state prior to stimulation. (**A**) Average chromatin accessibility at TSS of *dTrpA1*-induced ARGs in brains expressing *Elav-GAL4;UAS-dTrpA1* or *Elav-GAL4/CyO* (control) at 0 or 30 min of temperature shift. TSS is aligned to 0 at x-axis which indicates the distance from TSS in base pair. Negative numbers indicate upstream of TSS and positive indicates downstream. Same for all the graphs in this figure. (**B**) Average chromatin accessibility at TSS of *hr38* in brains expressing *Elav-GAL4;UAS-dTrpA1* or *Elav-GAL4/CyO* (control) at 0 or 30 min of temperature shift. (**C**) Average chromatin accessibility at TSS of heat-shock proteins in brains expressing *Elav-GAL4;UAS-dTrpA1* or *Elav-GAL4/CyO* (control) at 0 or 30 min of temperature shift. (**D**) Average chromatin accessibility at TSS of all expressed genes (FPKM > 0.1) and ARGs induced with different stimulation paradigms. Data are the average of all different conditions (genotypes and time points) and no changes of chromatin accessibility were observed among these conditions.

The following figure supplements are available for figure 6:

**Figure supplement 1.** Promoter regions of all annotated genes are more accessible in brains.

**Figure supplement 2.** Chromatin accessibility of TSS of *CG7995* show no changes with firing.

**Figure supplement 3.** Expression levels of ARGs at baseline levels.

inserted into a vector containing *mCherry* and *luciferase*, which originated from the Yin lab (*Figure 7A*) (*Tanenhaus et al., 2012*), and transgenic flies generated.

To assay the reporters, they were crossed to *Elav-GAL4;UAS-flp;UAS-ChR2-XXL*. In progeny expressing all the transgenes, GAL4 is expressed pan-neuronally and activates expression of *ChR2-XXL* to fire neurons. GAL4 also activates *flp*, which removes sequences between the two FRT sites in the reporter; this allows brain-specific expression of luciferase under the control of the enhancer, the transcription factor-multimer, and the promoter (*Figure 7A*). Adult progeny were subjected to a protocol like in *Figure 1*: adult flies were subjected to a single 30 s blue LED exposure, after which their luciferase levels were monitored every 10–15 min for several hours with a top counter (*Brandes et al., 1996*). All flies expressing *ChR2-XXL* showed seizure behavior within seconds of LED exposure as described above, whereas the control flies without *ChR2-XXL* expression were unaffected.

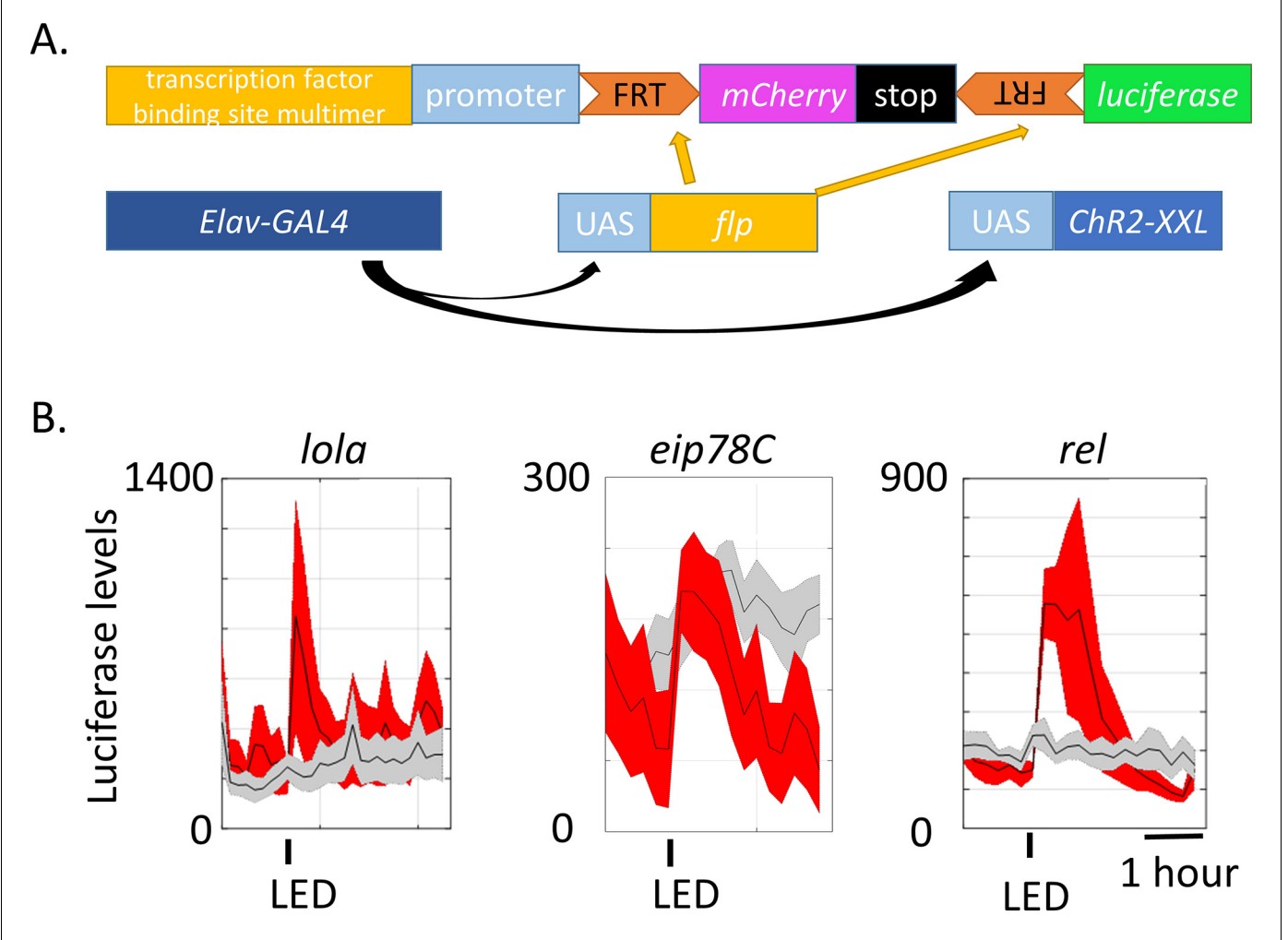

**Figure 7.** Generation of luciferase reporters for in vivo monitoring of neuronal activity. (**A**) Schemes of transgenes constructed for monitoring neuronal activity (top) and other transgenes crossed to be expressed in the same flies. Transcription factor binding sites include binding sites of LOLA, EIP78C and REL. Three of each transcription factor binding site were multimerized and fused upstream to the promoter and individually injected to fly embryos. These reporter-containing flies were then crossed to the existing *Elav-GAL4, UAS-FLP* and *UAS-ChR-XXL* (not in controls) flies. (**B**) Luciferase levels of flies expressing the constructed reporters exposed to a 30 s LED stimulation recorded every 10–15 min for ~4 hr. Gray indicates luciferase levels of control flies expressing all the transgenes in (**A**) except for *UAS-ChR2-XXL*. Red indicates flies expressing all the transgenes in (**A**). The experimental and control flies were put in the same 96-well plate and stimulated and recorded in parallel. The 30 s LED exposure is indicated by the black bars below the graphs. N = 8 flies. Representative images of multiple biological replicates are shown. Shaded regions represent ±SD.

The following source data and figure supplement are available for figure 7:

**Source data 1.** Motif enrichment in the 5 kb upstream regions of *ChR2-XXL*-induced ARGs.
**Source data 2.** Motif enrichment in the 1 kb upstream regions of *ChR2-XXL*-induced ARGs.
**Figure supplement 1.** Luciferase reporters for *br* and *cf2* show no change with LED stimulation in both experiment (red) and control (gray) flies.

The three transcription factors (LOLA, EIP78C, and REL) with high significance showed a dramatic increase in luciferase signal immediately after LED exposure, and the signal decayed in about an hour (*Figure 7B*). In contrast, the two transcription factors with much lower significance (*br, cf2*) showed no change (*Figure 7—figure supplement 1*.). Although the luciferase levels are somewhat noisy, they show that the three transcription factors with high significance are rapidly activated in response to firing. We also note that the different reporters may be activated in different neurons, i.

e., the results do not imply that we have identified a *fos*-eqivalent for *Drosophila*. However, they do indicate that the strategy is sound and should work for other activity-activated transcription factors and for the design of additional reporters, including for cell-specific neuronal activity (*Guo et al., 2016*).

## Discussion

Using high-throughput sequencing, we identified ARGs in fly brains in response to *ChR2-XXL*, *dTrpA1* and KCl-induced neural firing. Each stimulation paradigm induced expression of about 100 genes, and there is substantial overlap between the 3 sets of ARGs (*Figure 3D*). The most robust ARGs are the two known genes identified previously, *hr38* and *sr*; they are induced dramatically by *ChR2-XXL* and *dTrpA1* (*Figure 3E*). They are also induced with KCl albeit with smaller amplitudes (*Figure 3E*). 10 additional genes are also induced by all three protocols, and even more genes are induced by at least two paradigms (*Figure 3D*). Although the results dramatically increase the number of common ARGs, there are also a large number of specialized ARG candidate genes (*Figure 3D*). Not surprisingly perhaps, KCl has the most different ARGs. This is presumably because it is the only in vitro stimulation paradigm and stimulates the whole brain including non-neuronal cells and tissues.

We also used the *dTrpA1* paradigm to identify ARGs in sorted neurons. Many of them are specific to neuron type, and even the most robust brain ARGs are not necessarily induced by firing in every region of the brain (*Figure 5*, *Figure 5—figure supplement 1*). This is in contrast to heat-induced genes, most of which are shared by the three sources of RNA (*Figure 5—figure supplement 2*.).

Heat-induced gene expression unfortunately creates a complicated situation: some putative specific ARGs are also induced with the 18°C to 29°C shift in control (no *dTrpA1*) samples. These genes are thus excluded from the ARG list because they are defined as heat-induced by the cutoffs used in the analysis. Because it is possible that the temperature shift can lead to neuronal firing, e.g., by activating endogenous *dTrpA1*, there is currently no clear distinction between heat- and firing-induced genes. Future experiments with other tools like *ChR2-XXL* may be able to provide more information. In any case, the neuron-specific ARGs probably contribute to different physiological responses, which contribute in turn to distinct behavioral responses (*Guo et al., 2014*; *Ueno et al., 2012*). The results also provide good candidates as neural activity markers, e.g., *Figure 7*, and suggest that multiple genes should be tried, to accommodate cell type as well as firing heterogeneity.

Although the in vivo nature of our most important stimulation paradigms precluded using inhibitors to assay the dependence of ARGs on de novo protein synthesis (data not shown), the induction time scales of these fly ARGs, 60–90 min, resemble those of mammalian IEGs. A comparison between fly ARGs with mammalian IEGs therefore seems appropriate.

Whereas mammalian IEGs are highly enriched in transcription factors (*Spiegel et al., 2014*), they are less enriched in the fly brain and sorted neuron ARGs: *ChR2-XXL* induced ARGs contain only seven transcription factors among 96 genes; this is a much smaller fraction than in mammals (*Figure 1D*). KCl-induced ARGs show no enrichment in functions, and *dTrpA1*-induced ARGs in the cell types have a weak enrichment in different functions, such as steroid hormone receptor activity in brains, imaginal disc growth factor receptor binding in DA neurons and organic anion transmembrane transporter activity in PDF+ neurons (*Table 1*). The weak enrichment suggests diverse ARG functions, which resemble more closely mammalian DRGs or SRGs rather than PRGs (see below).

Less transcription factor activity and more diverse functions are consistent with the fact that fly homologues of the two most universal mammalian IEGs *c-fos* and *c-jun*, *kayak* and *jra* respectively, are not induced or poorly induced under most conditions. This is especially true for *kayak*; *jra* is

**Table 1.** Gene ontology analysis on *dTrpA1*-induced ARGs in different tissues with GOrilla.

|  | Functions | p-value | Genes |
|---|---|---|---|
| Brains | Steroid hormone receptor activity | 0.000536 | 3 |
| DA | Imaginal disc growth factor receptor binding | 0.000843 | 4 |
| PDF+ | Organic anion transmembrane transporter activity | 0.000742 | 5 |

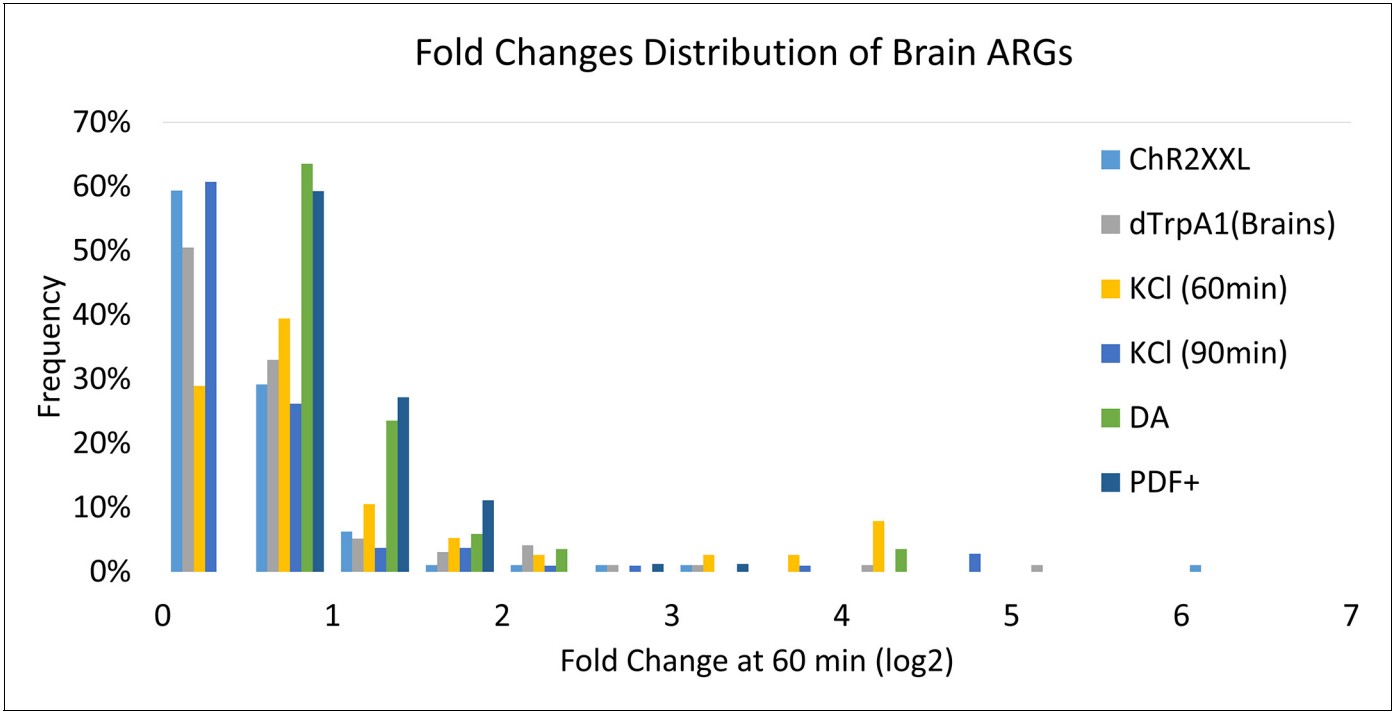

**Figure 8.** Fold changes distribution of all ARGs. X-axis indicates fold changes (log2) at 60 or 90 (for KCl) min. Y-axis indicates the frequency of genes. ChR2-XXL indicates *ChR2-XXL*-induced ARGs. dTrpA1 (brains) indicates *dTrpA1*-induced genes in brains. KCl (60 min/90 min) indicates KCl-induced ARGs in 60 or 90 min. DA indicates *dTrpA1*-induced ARGs in DA neurons. PDF+ indicates *dTrpA1*-induced ARGs in PDF+ neurons.

induced by both *dTrpA1* and KCl in brain but with quite small fold changes (*Figure 3—source data 1* and *2*).

We also noticed that most fly ARGs show rather small fold-changes upon stimulation (*Figure 8*). It is unlikely that this is due to brain tissue heterogeneity as similar results were obtained with the sorted neurons. A similar weak induction of seizure-induced genes was previously reported in fly heads (*Guan et al., 2005*). As only the first 60–90 min after stimulation was assayed, it is possible that the small fold-changes reflects a slower induction time course in flies than in mammals. Another possibility is that the nature of the firing, e.g., firing rate, has an impact on the induction amplitudes, time courses and even the genes induced. The differences between the three stimulation paradigms (*Figure 3*) indicate that this possibility may have some merit. Nonetheless and taken together with the stimulation-paradigm and cell type-specific induction of ARGs, the modest fold-changes at 60–90 min helps explain the long-standing failure to identify *bona fide* and well-accepted ARGs in flies.

Instead of having shorter average gene lengths like in mammals, fly ARGs (brain as well as sorted neurons) show even longer gene length distributions than average fly genes (*Figure 2C*, *Figure 9*). Several robust ARGs are above 10 kb with large first introns (*Figure 2—figure supplement 1*). For example, *hr38* is ~31 kb and shares high protein sequence identity with the mammalian protein encoded by *Nr4a2*; it spans only ~16.8 kb. *sr*, the shorter isoform of which is induced by firing, spans ~11.7 kb, whereas EGR3, the mouse gene that encodes a protein with high identity to SR, is only ~5.2 kb in length. Considering the much smaller genome size and gene size of flies compared to mammals and the potential slower transcription elongation rates in flies, the long gene lengths and large first introns of fly ARGs may counterintuitively serve to significantly slow down induction rates (*Ardehali and Lis, 2009*). Another possibility is that the large gene size reflects important regulatory elements for firing-mediated transcription embedded in the introns of these genes. Perhaps they are relevant to the diversity and cell type specificity of most fly ARGs.

From this point of view, fly ARGs resemble more closely mammalian DRGs or even SRGs than IEGs (*Spiegel et al., 2014*; *Tullai et al., 2007*). Mammalian IEGs are largely shared between different types of neurons such as excitatory and inhibitory neurons (*Spiegel et al., 2014*), but fly ARGs like

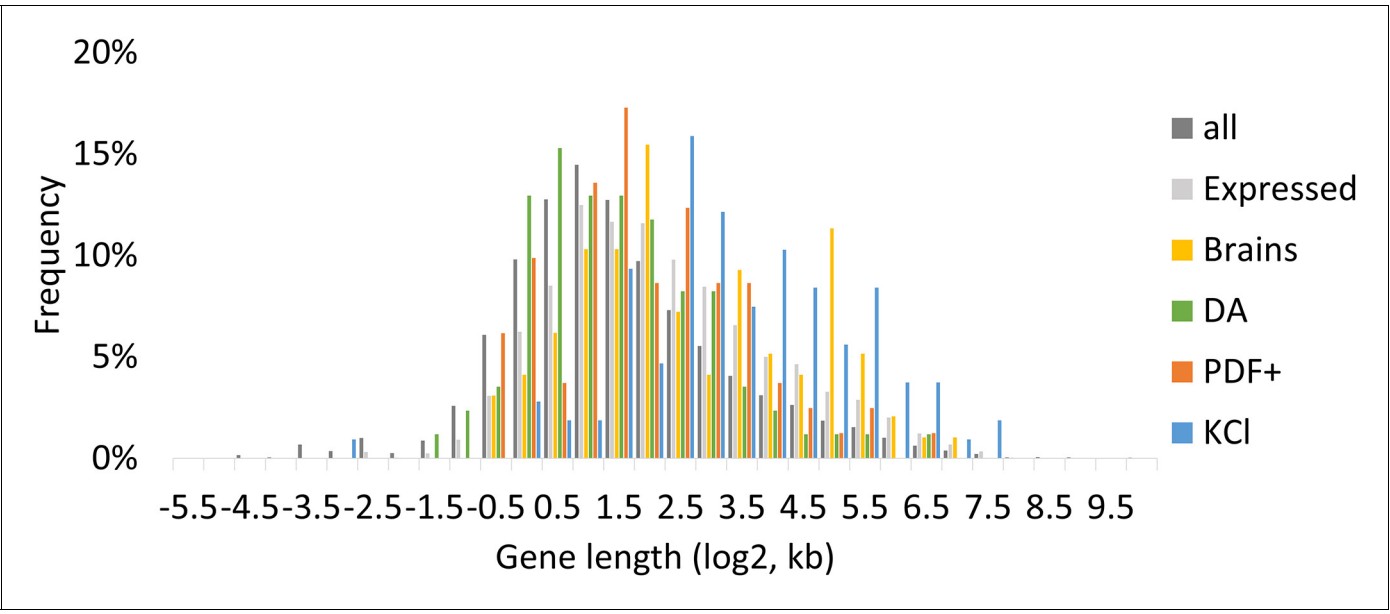

**Figure 9.** Gene length distribution of all annotated genes and ARGs. All indicates all annotated genes. Expressed indicates genes with FPKM > 0 at baseline, FPKM > 10 with stimulation in brains. Brains indicate *dTrpA1*-induced ARGs in brains. DA indicates *dTrpA1*-induced ARGs in DA neurons. PDF + indicates *dTrpA1*-induced ARGs in PDF+ neurons. KCl indicates KCl-induced ARGs in 90 min.

SRGs appear quite tissue-type specific, i.e., many genes are only induced in one of the tissue types tested despite similar baseline levels (*Figure 5*) (*Mardinly et al., 2016*). This is in line with the fact that fly neurons appear quite heterogeneous, e.g., they have distinct transcript enrichment and electrophysiological properties (*Liang et al., 2016*; *Nagoshi et al., 2010*). Although unlikely in our view, it remains possible that PDF+ and DA neurons are outliers and that ARGs in most other neurons are less cell type-specific. Nonetheless, we hypothesize that specific ARGs are an important feature of neuronal identity and play a role in neuron-specific physiological properties. As an example, preliminary analysis indicates that the PDF cell-ARGs are a subset of the mRNAs that undergo circadian oscillations within these neurons ([*Kula-Eversole et al., 2010*] and data not shown).

The non-identical ARGs with the different paradigms also suggest different mechanisms for responding to different modes of stimulation. A less stereotyped response may be generally characteristic of invertebrates. Even non-mammalian vertebrates may be quantitatively if not qualitatively different from mammals. For example, induction of IEGs like *fos* and *egr1* was also observed in birds and zebrafish but with smaller increases (about 4 to 6-fold) than human and mice (usually well above 10-fold) (*Bébien et al., 2003*; *Burmeister and Fernald, 2005*; *Lopes et al., 2015*; *Mokin and Keifer, 2005*; *Moore and Whitmore, 2014*). However, more high-throughput data are required to determine if the non-mammalian vertebrates resemble more closely flies or mammals. In addition, different stimulation paradigms were employed in the non-mammalian vertebrate studies, making the data less comparable. In any case, one can imagine that mammals need a more potent initial transcriptional system than flies to respond to firing. Not only are mammalian neurons bigger with more and longer processes, but they also have much larger genomes (longer genomic DNA and more enhancers to bind); both of these factors may require higher levels of transcripts to generate a required higher abundance of transcription factors. The increase may be required to activate the thousands of enhancers in mammals rather than the hundreds in flies, and/or more structural proteins are needed to change the morphology of 'giant' mammalian neurons.

Nonetheless, one feature shared by fly ARGs and mammalian IEGs is the relatively permissive state of the TSS chromatin structure (*Fowler et al., 2011*). ATAC-seq shows that the chromatin accessibility of the ARG TSS is higher than that of other genes (*Figure 6D*), and chromatin architecture contributes to gene expression regulation (*Wu, 1997*). Although no further opening is observed with firing, it is possible that more subtle changes such as histone modifications occur and are not

detectable with ATAC-seq (*Figure 6A*). In the cases of mammals, histones around IEGs promoters are modified to be more accessible to transcription factors at baseline; the modification levels could be even higher with firing (*Kim et al., 2010*). In any case, the data taken together suggest some blurring of the distinction between IEGs and SRGs in fly ARGs: open chromatin and rapid induction but a diverse set of neuron-specific functions.

Finally, we successfully generated luciferase reporters for in vivo neural activity monitoring based on the *ChR2-XXL* data (*Figure 7*). Rather than using the regulatory region of a single gene to construct a reporter, we generalized a previous strategy (*Tanenhaus et al., 2012*) and used enriched transcription factor binding sites combined with FRT sites for cell type specific expression and a short-lived luciferase gene. This was not only to amplify the signal but also to avoid the situation in which a particular promoter is only responsive in certain neurons or to certain types of stimulation. We thought this was likely considering that ARGs are stimulation paradigm- and cell type-specific. Indeed, we tried many Janelia Research Campus GAL4 drivers, which use different regions of the *hr38, sr* and *cbt* promoters for neural activity monitoring, but they were all either unresponsive or generated very low luciferase levels (data not shown).

Two of the three positive reporters (*Figure 7*) work well for circadian monitoring of neuronal activity in PDF+ neurons (data not shown), suggesting the signal to noise ratio is sufficient for monitoring small subsets of neurons. This result and others indicate that our strategy as well as these specific reporters can provide the temporal and spatial resolution for in vivo monitoring of small numbers of specific neurons. The strategy also builds a foundation for future elaboration and optimization.

## Materials and methods

### *Drosophila* stocks

*Drosophila* (RRID:FlyBase_FBst1021211) were reared on standard cornmeal/agar medium supplemented with yeast under 12 hr LD cycles at 25°C. *Elav-GAL4* (C155) was obtained from Bloomington Stock Center. *UAS-ChR2-XXL* was a kind gift from Kittel Lab. *UAS-dTrpA1* was from Dr. Paul Garrity. *TH-GAL4* was described in (*Guo et al., 2014*). *Pdf-GAL4* was described in (*Stoleru et al., 2004*). *UAS-mCD8GFP;Pdf-GAL4* was described in (*Nagoshi et al., 2010*). Flies expressing both *GAL4* and *UAS-dTrpA1* were reared at 18°C until tested. *UAS-FLP* (#55804) was obtained from Bloomington Stock Center.

### Stimulation protocols

For the *ChR2-XXL* paradigm, both the control and experimental flies were transferred from normal fly food to all trans-retinal (ATR) – containing food for one day and kept in dark until stimulation. ATR powder was dissolved in EtOH into 100 mM stock, and further diluted to 0.4 mM as final working concentration (*Klapoetke et al., 2014*). To fire neurons, 30 s 10 Hz 0.7 V blue LED with 5 ms pulse width was given to flies. Flies were allowed to recover in dark for 15, 30, 60 min.

For the *dTrpA1* paradigm, both the control and experimental flies were reared at 18°C and shifted to 29°C for 0, 15, 30, 60 min to fire neurons.

For the KCl paradigm, Canton-S WT fly brains were dissected in adult hemolymph-like medium (AHL) consisting 108 mM NaCl, 5 mM KCl, 2 mM $CaCl_2$, 8.2 mM $MgCl_2$, 4 mM $NaHCO_3$, 1 mM $NaH_2PO_4$, 5 mM trehalose, 10 mM sucrose and 5 mM HEPES and transferred to depolarization buffer containing 28 mM NaCl, 85 mM KCl, 2 mM $CaCl_2$, 8.2 mM $MgCl_2$, 4 mM $NaHCO_3$, 1 mM $NaH_2PO_4$, 5 mM trehalose, 10 mM sucrose and 5 mM HEPES (*Shang et al., 2013*; *Wang et al., 2003*). pH for both buffers was adjusted to 7.5, and mOsm to 265.

For *ChR2-XXL* and *dTrpA1* stimulation performed under ex vivo condition, fly brains were dissected in AHL and subjected to either a 30 s LED exposure or a 60 min 29°C water bath incubation.

For all experiments assayed from brains and heads, 8–10 brains (50% males and females) from ~7 day-old young adults were used for each condition. Brains were carefully dissected to remove all non-brain tissues, and only intact undamaged brains were used for RNA extraction. For sorted neurons, 20 young males and 20 young females were used.

## ArcLight imaging

Fly brains were dissected in AHL and then mobilized using a pin anchored to the chamber bottom laid with Sylgard (Dow Corning, Midland, MI) (*Shang et al., 2013*; *Wang et al., 2003*). Depolarization buffer was perfused into the chamber using a gravity-fed ValveLink perfusion system (Automate Scientific, Berkeley, CA) (*Haynes et al., 2015*; *Shang et al., 2013*). Imaging was performed with an Olympus BX51WI fluorescence microscope (Olympus, Center Valley, PA) under an Olympus x60 (0.90W, LUMPlanFI) water-immersion objective, and captured using a charge-coupled device camera (Hamamatsu ORCA C472-80-12AG). The following filter sets were used for excitation and emission (Chroma Technology, Bellows Falls, VT): excitation, HQ470/x40; dichroic, Q495LP; emission, HQ525/50m. μManager was used for recording with 2 Hz with 4 x binning with 500 ms exposure time and 50 ms intervals (*Edelstein et al., 2010*).

## RNA extraction, amplification for sequencing libraries and qPCR validation

Fly brains or heads were isolated for RNA extraction using TRIzol. mRNA libraries were constructed following Illumina Trueseq protocol. Nascent RNA libraries were constructed as described in (*Rodriguez et al., 2013*). mRNA libraries from sorted neurons were prepared as described in (*Abruzzi et al., 2015*). ATAC-seq libraries were done using Nextera DNA Sample Prep Kit. Libraries were sequenced using HiSeq, NextSeq or MiSeq sequencers.

For qPCR validation, total RNA was reverse-transcribed with random primer (Promega, C1181) and gene expression was then quantified with SYBR Green PCR Master Mix using primers listed in *Supplementary file 1*.

## Sequencing data analysis

RNA libraries were mapped to *Drosophila* genome dm3 using Tophat (*Trapnell et al., 2009*) with default setting (e.g. two mismatches allowed) and expression levels were quantified using Cufflinks (*Trapnell et al., 2012*). FPKM > 0 for un-stimulated groups and FPKM > 10 for stimulated groups were used as expression cutoff. HTSeq (*Anders et al., 2015*) was used to produce raw read count for each genes. EdgeR (*McCarthy et al., 2012*; *Robinson et al., 2010*; *Robinson and Smyth, 2007*, *2008*; *Zhou et al., 2014*) was used for statistical analysis. N = 3 biological replicates for libraries from brains. N = 3–4 biological replicates for libraries from sorted neurons. p-value < 0.01 was defined as significance except for PDF+ ARGs (p-value < 0.05 due to slightly bigger variation). To distinguish heat- or firing-induced ARGs in *dTrpA1* paradigms, t-test was used among replicates between the experiment and control flies. ARGs are defined to be firing-induced only when P-value < 0.05 and experiment flies show bigger induction than control flies. For ATAC-seq, Trim Galore (http://www.bioinformatics.babraham.ac.uk/projects/trim_galore/) is used to trim low quality bases (quality score < 20) on either end of the reads and remove adapter/tagmentation sequences. Bowtie2 (*Langmead and Salzberg, 2012*) was used to map the reads to reference genome with parameters '-D 20 -R 3 -N 1 -L 20'. Alignments below a score of 10 were discarded, and PCR duplicates were removed using Picard (version 1.119). Chromatin accessibility at TSS were pooled and averaged as described in (*Menet et al., 2014*).

## Generation of luciferase reporter flies

The cre-luc vector was a kind gift from Jerry Yin lab. The cre sites of the vector were swapped with binding site multimers (3X) of transcription factors including LOLA, EIP78C, REL, BR and CF2 using the primers in *Supplementary file 1*. Transgenes were injected into w1118 flies with the P-element transformation service of BestGene Inc.

## Acknowledgements

We thank Drs. Oliver Hobert and Haosheng Sun from Columbia University for comments on an early version of the manuscript. We also thank Drs. Suzanne Paradis, Michael Marr, Nelson Lau, Patrick Emery and Leslie Griffith, for comments on a near-final version. Many thanks go to members of Rosbash lab for thoughtful discussion, and especially Kate Abruzzi for comments.

# Additional information

## Funding

| Funder | Author |
|---|---|
| Howard Hughes Medical Institute | Xiao Chen<br>Reazur Rahman<br>Fang Guo<br>Michael Rosbash |

The funders had no role in study design, data collection and interpretation, or the decision to submit the work for publication.

## Author contributions

XC, Conception and design, Acquisition of data, Analysis and interpretation of data, Drafting or revising the article; RR, Analysis and interpretation of data, Drafting or revising the article; FG, Provided advice and help, Conception and design; MR, Conception and design, Analysis and interpretation of data, Drafting or revising the article

## Author ORCIDs

Michael Rosbash, http://orcid.org/0000-0003-3366-1780

# Additional files

## Supplementary files

• Supplementary file 1. Supplementary tables. (A) Primers for qPCR validation. (B) Primers for reporter generation

## Major datasets

The following dataset was generated:

| Author(s) | Year | Dataset title | Dataset URL | Database, license, and accessibility information |
|---|---|---|---|---|
| Xiao Chen | 2016 | Genome-wide Identification of Neuronal Activity-regulated Genes in Drosophila | https://www.ncbi.nlm.nih.gov/geo/query/acc.cgi?acc=GSE83976 | Publicly available at NCBI Gene Expression Omnibus (accession no: GSE83976) |

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
