## [Decision Letter]

Thank you for submitting your article "Genome-wide Identification of Neuronal Activity-regulated Genes in *Drosophila*" for consideration by *eLife*. Your article has been favorably evaluated by a Senior Editor and two reviewers, one of whom, Hugo Bellen, is a member of our Board of Reviewing Editors.

The reviewers have discussed the reviews with one another and the Reviewing Editor has drafted this decision to help you prepare a revised submission.

Chen et al. used three different neuronal stimulation methods and RNAseq to systematically identify Activity-Regulated Genes (ARGs) in *Drosophila*. The study identified 12 ARGs (including previously known ARGs, hr38 and striped) that overlap in the three paradigms but also revealed a large number of stimuli-specific ARGs, suggesting that neurons may respond differently to different stimuli. The authors further performed RNAseq on small groups of neurons isolated by cell sorting (Dopaminergic and Pdf neurons, stimulated by the dTrpA1-strategy) and found that although there were some overlaps, many genes were cell type specific. In addition, the authors investigated the chromatin accessibility of the ARGs pre- and post-stimulation and found that in contrast to mammalian IEGs that show some changes in chromatin structure upon stimulation, the fly ARGs are in an open chromatin state to begin with. Together with the findings that fly ARGs are not enriched for transcription factors and small genes, the authors find an interesting difference between mammalian IEGs and *Drosophila* ARGs, and conclude that fly ARGs are more similar to mammalian secondary response genes (SRGs) or delayed response genes (DRGs). Finally, using the information from these experiments, the authors generated five Luciferase-based reporters and show that three of them show some stimulation induced activation in vivo.

Overall, we feel that the manuscript is well conceived and well written. Identification of new ARGs and showing a proof-of-principal that this information can be used to generate new tools is valuable. In addition, pan-neuronal and subtype specific transcriptomic profiles of neurons treated with different stimulations are also unique and interesting datasets that can be further mined to extract interesting biological information. The finding that, in *Drosophila*, activity-induced transcription happens downstream of the level of chromatin accessibility will also allow researchers to begin to study the similarities and differences in the mechanism of activity-dependent gene activation between different model organisms. Finally, although not optimized yet, construction of several luciferase based neuronal activity reporter further adds value to this paper. As such, it is likely to be of broad interest and appropriate for publication in *eLife*; however, there are a few issues that must first be addressed:

Essential revisions (requiring experimental studies):

1) In all of the figures regarding expression profiling, there are no data confirming that the primary findings in RNAseq can be verified by other methods such as qPCR. We assume that the RNAseq was performed on biological replicates (although we cannot find such info in the Materials and methods). Even so we feel that having independent sets of data to backup these primary findings are critical. The authors should take several genes (top candidates from each of the stimulation paradigms that are shared and are unique) and perform qPCR or other assays to detect RNA transcripts (or protein if there is a change) to demonstrate that their findings are reproducible and can be verified by independent methods.

2) The authors find that although there are overlapping genes, ARGs found in different stimulation paradigms are significantly different. The authors speculate this could be reflecting some gene expression changes in non-neuronal cells (glia etc.), global changes that happen independent of neuronal activation (light-response, heat-shock response) or the difference in in-vivo versus ex-vivo experiments. One possibility that the authors do not discuss is the strength and duration of the stimulation. Based on the Methods section, the ChR2 paradigm stimulates the flies only for 30 seconds at the beginning and follow transcriptional changes, while dTrpA1 and high K^+^ paradigms continuously activate the neurons for the entire duration (up to 60/90min). If the authors perform a longer stimulation in ChR2 flies, they may see more genes that overlap with the other 2 paradigms. Also, in order to determine if the difference between high-K^+^ stimulation and the other two paradigms are simply due to differences in-vivo and ex-vivo sample preparation, the authors can perform an ex-vivo experiment for the ChR2 and dTRPA1 flies as well and see how that may affect the list of ARGs. Considering that additional RNAseq may require significant time, effort, and cost, the authors can select several high-confident hits from each of the paradigms (see major point 1) and test them via qPCRs or other methods.

3) In Figure 7, the authors show that 3 reporters they generated respond to ChR2-mediated neuronal activation. However, they do not show how these reporters behave in the other two stimulation paradigms (dTrp1A, high K^+^). Was there a reason the author only focused on ChR2-ARGs? Is the binding site of Lola, Eip78 and Relish enriched in ARG genes activated by the other two paradigms? How do these reporters (and the two that didn't respond to ChR2-activation) behave when stimulated with dTrp1A and high-K^+^? Since a key conclusion that the authors draw from their RNAseq data is that stimulation paradigms make a big difference, the authors should test their reporter under different stimulation conditions. In addition, the authors should comment on whether their reporter signal is strong enough that they can be used to visualize neuronal activity in all (e.g. elav-GAL4) or subpopulation (e.g. TH-GAL4, Pdf-GAL4) of neurons.

Additional revisions (Textural changes/clarification):

1) Do any of the ARG genes the authors identified have known roles in neuronal plasticity? If so, what portion? This should be included in the Discussion.

2) We cannot find the information of how many flies or brains were used for each experiment. Also, were all of the flies of the same sex or a mix? How old were the flies when these experiments were performed? The authors should provide enough information so that others can replicate the findings.

3) For ChR2-flies, the authors state "A 30-second 10 Hz LED exposure was sufficient to induce a uniform seizure within seconds, and all flies were able to recover within 15 min." For the dTrpA1 experiments, however, no phenotypic descriptions are provided. Did the dTrpA1 flies also show a similar seizure phenotype? If the flies do not show seizures or show a different behavioral defect during the stimulation, the two stimuli maybe acting on different neuronal populations (despite the two proteins being expressed pan-neuronally with elav-GAL4) leading to a difference in the ARG list.

4) The authors should state the amount/concentration of ATR that was fed to theChR2 flies.

5) In Figure 2, Is the 'weak anti-correlation' between change in gene expression and relative gene length significant? What is the R value?

6) In Figure 7, there appears to be a second peak of luciferase activity at the end of recording with the eip78C reporter. Is this typical? Background? Non-specific activation? Is the signal observed significantly different from the control (looks very similar)? Additionally, in Figure 7—figure supplement 1, both of these 'control' reporters seem to have less of a response compared to controls. Is there any possible explanation for that phenomenon? Is the difference significant?

7) Many of the experiments used a control with CyO in the background. While we understand that there are advantages to using siblings for comparison, balancer chromosomes often incur unpredictable phenotypes. Therefore, were any steps taken to ensure that the CyO chromosome was not significantly different from WT?

8) In the second paragraph of the subsection “Promoter Regions of ARGs are at Permissive State Prior to Stimulation”, what do you mean when you refer to "all annotated genes"? This phrasing is used several times throughout the manuscript, but is not adequately explained. Does this mean all fly genes, all ARGs or something else?

9) Figure 3: Misalignment of "Dissecte d Brains" in A.

10) In Figure 3, were the 8 genes shown in E selected at random or was there a reason to leave out the other 4?

11) Figure 4: The axes for A and B are not internally consistent; I believe that all coordinates should range from -5 to 5.

12) Formatting of Tables 9 and 10 needs to be adjusted. Also, since only 3 of the columns in both of these tables have different values, the tables could be shrunk down with the rest of the information (which is the same for all columns) described in a legend.

---

## [Author Response]

*Essential revisions (requiring experimental studies):*

*1) In all of the figures regarding expression profiling, there are no data confirming that the primary findings in RNAseq can be verified by other methods such as qPCR. We assume that the RNAseq was performed on biological replicates (although we cannot find such info in the Materials and methods). Even so we feel that having independent sets of data to backup these primary findings are critical. The authors should take several genes (top candidates from each of the stimulation paradigms that are shared and are unique) and perform qPCR or other assays to detect RNA transcripts (or protein if there is a change) to demonstrate that their findings are reproducible and can be verified by independent methods.*

Sorry for the confusion. Yes, all the ‘N’ mentioned in the manuscript indicates biological replicates. We’ve added this information in the both the text, legends and Methods sections.

To respond to your request, RT-qPCR has now been performed for verification purposes, and most of the genes tested are consistent with the sequencing results (Figure 3—figure supplement 2).

*2) The authors find that although there are overlapping genes, ARGs found in different stimulation paradigms are significantly different. The authors speculate this could be reflecting some gene expression changes in non-neuronal cells (glia etc.), global changes that happen independent of neuronal activation (light-response, heat-shock response) or the difference in in-vivo versus ex-vivo experiments. One possibility that the authors do not discuss is the strength and duration of the stimulation. Based on the Methods section, the ChR2 paradigm stimulates the flies only for 30 seconds at the beginning and follow transcriptional changes, while dTrpA1 and high K^+^ paradigms continuously activate the neurons for the entire duration (up to 60/90min). If the authors perform a longer stimulation in ChR2 flies, they may see more genes that overlap with the other 2 paradigms. Also, in order to determine if the difference between high-K^+^ stimulation and the other two paradigms are simply due to differences in-vivo and ex-vivo sample preparation, the authors can perform an ex-vivo experiment for the ChR2 and dTRPA1 flies as well and see how that may affect the list of ARGs. Considering that additional RNAseq may require significant time, effort, and cost, the authors can select several high-confident hits from each of the paradigms (see major point 1) and test them via qPCRs or other methods.*

In response to these suggestions, we performed several experiments and described most of the results in a single paragraph in the text and added the figures as Figure 3—figure supplement 3 and Figure 3—figure supplement 4. We added this detail, text and figures, so that the reviewers and editor can examine the results. However, we find the addition rather cumbersome. So we are happy to refer to these PCR data as data not shown if editor/reviewers agree. On the other hand, we are more than willing to keep all this information in the manuscript. This is your choice.

*3) In Figure 7, the authors show that 3 reporters they generated respond to ChR2-mediated neuronal activation. However, they do not show how these reporters behave in the other two stimulation paradigms (dTrp1A, high K^+^). Was there a reason the author only focused on ChR2-ARGs? Is the binding site of Lola, Eip78 and Relish enriched in ARG genes activated by the other two paradigms? How do these reporters (and the two that didn't respond to ChR2-activation) behave when stimulated with dTrp1A and high-K^+^? Since a key conclusion that the authors draw from their RNAseq data is that stimulation paradigms make a big difference, the authors should test their reporter under different stimulation conditions. In addition, the authors should comment on whether their reporter signal is strong enough that they can be used to visualize neuronal activity in all (e.g. elav-GAL4) or subpopulation (e.g. TH-GAL4, Pdf-GAL4) of neurons.*

The reporters were originally generated based on the bioinformatics on the ChR2-XXL induced ARG list, and therefore tested only with that same paradigm. The ChR2-ARGs were chosen because we think this is a more physiological way to fire neurons than the other two paradigms, which involved heat and high K^+^ depolarization. For dTrpA1 at least, the data back this up with lots of heat shock genes induced by the temperature shift even without dTrpA1. We later found that the 3 positive transcription factors are also enriched for the other two stimulation paradigms. We didn’t test the reporters by monitoring ex vivo brains with the KCl paradigm. First, we are somewhat fixated on awake-behaving flies as we are developing these reporters and currently using them as a real time monitoring tool for living animals (see below). In addition, and from a more practical point of view, our top counter machine although very sensitive is not well-adapted to ex vivo monitoring.

In contrast and also as suggested, we tried the reporters with the TrpA1 paradigm (Experimental flies: Elav-GAL4;UAS-dTrpA1;UAS-FLP;reporters, control flies: Elav-GAL4;UAS-FLP;reporters). Somewhat surprisingly perhaps, all reporters including the two negative ones responded to the 21°C to 30°C temperature shift, without UAS-dTrpA1 as well as with (data not shown), suggesting that heat may increase the activity of many transcription factors. Although heat can also fire some neurons, the induction here is more likely due to heat alone since the levels of induction are indistinguishable between the experimental and control flies for all reporters. The observation also suggests that these reporters are not exclusively responsive to neuronal firing. However, we cannot at present exclude the possibility that this is an artifact, for example a temperature effect on luciferase reporter activity. We cannot sort this out within the time frame of returning the manuscript, so the assay of the reporters remains restricted to ChR2 in this paper, where fortunately we have convincing negative controls.

On the other hand, the three reporters have been tested under the control of Pdf-GAL4 since submitting this manuscript. We could not easily combine them with ChR2 in the PDF cells for complicated genetic reasons. However, a clear circadian pattern can be observed for both rel and lola but not eip78C, suggesting low EIP78C levels or activity in PDF+ cells. Importantly, the pattern is indistinguishable from our in vivo Ca++ reporter system (Guo et al. 2016). This indicates that the reporters “work” in small numbers of neurons and even respond to more natural firing conditions. These results will be reported soon in experiments that describe the firing patterns of different circadian neurons. We now make a bare bones “data not shown” statement in the revised manuscript to indicate that the reporters work in PDF neurons.

*Additional revisions (Textural changes/clarification):*

*1) Do any of the ARG genes the authors identified have known roles in neuronal plasticity? If so, what portion? This should be included in the Discussion.*

Many fly genes are involved in plasticity, with functions from regulating chromatin structure to post-translational modification. There are for sure some hits, but it is hard to say accurately what fraction they are.

*2) We cannot find the information of how many flies or brains were used for each experiment. Also, were all of the flies of the same sex or a mix? How old were the flies when these experiments were performed? The authors should provide enough information so that others can replicate the findings.*

The information is now added to the Methods section.

*3) For ChR2-flies, the authors state "A 30-second 10 Hz LED exposure was sufficient to induce a uniform seizure within seconds, and all flies were able to recover within 15 min." For the dTrpA1 experiments, however, no phenotypic descriptions are provided. Did the dTrpA1 flies also show a similar seizure phenotype? If the flies do not show seizures or show a different behavioral defect during the stimulation, the two stimuli maybe acting on different neuronal populations (despite the two proteins being expressed pan-neuronally with elav-GAL4) leading to a difference in the ARG list.*

The stimulation paradigm used for TrpA1 shifts the flies from 18°C to 29°C. No seizures were observed. However, shifting to an even higher temperature causes seizures, suggesting a similar or overlapping set of neurons are stimulated in both cases. However, we avoided shifting to above 30°C to minimize effects from heat and stress. Although it is possible that the TrpA1 paradigm is not as strong as the ChR2-XXL paradigm, similar levels of induction were observed for the genes in common between the two paradigms.

*4) The authors should state the amount/concentration of ATR that was fed to theChR2 flies.*

The concentration was and still is mentioned in the Methods section.

*5) In Figure 2, Is the 'weak anti-correlation' between change in gene expression and relative gene length significant? What is the R value?*

R^2^ for Figure 2 is 0.13, and for Figure 2 it is 0.03. Although.13 is low, it is still 4 times higher than.03. In addition, the low numbers suggest that gene length is not the only factor that determines induction rate. We suspect that the low numbers are also influenced by the many genes with low fold-changes. Nonetheless, we recognize that this figures is marginal and are of course willing to remove it if that is the reviewer/editorial consensus.

*6) In Figure 7, there appears to be a second peak of luciferase activity at the end of recording with the eip78C reporter. Is this typical? Background? Non-specific activation? Is the signal observed significantly different from the control (looks very similar)? Additionally, in Figure 7—figure supplement 1, both of these 'control' reporters seem to have less of a response compared to controls. Is there any possible explanation for that phenomenon? Is the difference significant?*

This is atypical, which was not described properly in the Figure legend. As mentioned (also added to the text), the reporters are a little noisy, which may indicate that they also respond to spontaneous firing. In addition, we have evidence that the reporters are regulated by other factors such as circadian rhythms (see above). We truncated the last time point of this graph in the revised manuscript so it looks less confusing. The 1^st^ time point with low values was generated by a program error in Figure 7—figure supplement 1, which is now also removed. Neither the control nor the experimental flies respond to LED stimulation (indicated by black bars beneath the graphs) in the updated figure.

*7) Many of the experiments used a control with CyO in the background. While we understand that there are advantages to using siblings for comparison, balancer chromosomes often incur unpredictable phenotypes. Therefore, were any steps taken to ensure that the CyO chromosome was not significantly different from WT?*

Gene expression of the CyO flies at baseline levels are similar to that of the TrpA1 flies, showing a very high R^2^. In response to heat shocks, they also show similar induction amplitudes. Lastly, the chromatin structure of CyO flies at baseline is also similar to the TrpA1 flies. Therefore, we believe that the CyO flies are not distinguishable from the gene expression point of view from our dTrpA1 flies.

*8) In the second paragraph of the subsection “Promoter Regions of ARGs are at Permissive State Prior to Stimulation”, what do you mean when you refer to "all annotated genes"? This phrasing is used several times throughout the manuscript, but is not adequately explained. Does this mean all fly genes, all ARGs or something else?*

It means all fly genes. These features were downloaded from Flybase. The link for download is added to the manuscript now.

*9) Figure 3: Misalignment of "Dissecte d Brains" in A.*

Fixed.

*10) In Figure 3, were the 8 genes shown in E selected at random or was there a reason to leave out the other 4?*

Sorry for missing the other 4 genes. They are now put back on.

*11) Figure 4: The axes for A and B are not internally consistent; I believe that all coordinates should range from -5 to 5.*

The axes are corrected. But since -5 to 5 makes the dots too clustered, we changed to -2 to 4.

*12) Formatting of Tables 9 and 10 needs to be adjusted. Also, since only 3 of the columns in both of these tables have different values, the tables could be shrunk down with the rest of the information (which is the same for all columns) described in a legend.*

Thank you for the advice. The tables have been shrunk down as recommended.